Redescription of Phymolepis  cuifengshanensis (Antiarcha: Yunnanolepididae) using high-resolution computed tomography and new insights into anatomical details of the endocranium in antiarchs

Wang Yajing
Zhu Min zhumin@ivpp.ac.cn
Key Laboratory of Vertebrate Evolution and Human Origins of Chinese Academy of Sciences, Institute of Vertebrate Paleontology and Paleoanthropology, Chinese Academy of Sciences , Beijing , China
CAS Center for Excellence in Life and Paleoenvironment , Beijing , China
University of Chinese Academy of Sciences , Beijing , China
De Baets Kenneth
Electronic publication date: 2018 May 28
Publication date: 2018
Volume: 6
Electronic Location ID: e4808
Received 2018 Feb 27; Accepted 2018 Apr 30
Copyright: ©2018 Wang and Zhu
Copyright year: 2018
Copyright holder: Wang and Zhu
License: This is an open access article distributed under the terms of the Creative Commons Attribution License, which permits unrestricted use, distribution, reproduction and adaptation in any medium and for any purpose provided that it is properly attributed. For attribution, the original author(s), title, publication source (PeerJ) and either DOI or URL of the article must be cited.
License URL: https://creativecommons.org/licenses/by/4.0/

Keywords: Devonian, Gnathostomes, Placoderms, CT scanning, Phylogeny, Endocranial morphology, Antiarchs, Anatomy

Funding: Strategic Priority Research Program of Chinese Academy of Sciences XDA19050102 Natural Science Foundation of China 41530102 Key Research Program of Frontier Sciences of CAS QYZDJ-SSW-DQC002 This research was funded by the Strategic Priority Research Program of Chinese Academy of Sciences (XDA19050102), Natural Science Foundation of China (41530102), Key Research Program of Frontier Sciences of CAS (QYZDJ-SSW-DQC002), and CAS Funds for Paleontology Fieldwork and Fossil Preparation. The funders had no role in study design, data collection and analysis, decision to publish, or preparation of the manuscript.

==============================
Background

Yunnanolepidoids constitute either the most basal consecutive segments or the most primitive clade of antiarchs, a highly diversified jawed vertebrate group from the Silurian and Early Devonian periods. Although the general morphology of yunnanolepidoids is well established, their endocranial features remain largely unclear, thus hindering our further understanding of antiarch evolution, and early gnathostome evolution. Phymolepis cuifengshanensis, a yunnanolepidoid from the Early Devonian of southwestern China, is re-described in detail to reveal the information on endocranial anatomy and additional morphological data of head and trunk shields.

Methods

We scanned the material of P. cuifengshanensis using high-resolution computed tomography and generated virtual restorations to show the internal morphology of its dermal shield. The dorsal aspect of endocranium in P. cuifengshanensis was therefore inferred. The phylogenetic analysis of antiarchs was conducted based on a revised and expanded dataset that incorporates 10 new cranial characters.

Results

The lateroventral fossa of trunk shield and Chang’s apparatus are three-dimensionally restored in P. cuifengshanensis. The canal that is positioned just anterior to the internal cavity of Chang’s apparatus probably corresponds to the rostrocaudal canal of euantiarchs. The endocranial morphology of P. cuifengshanensis corroborates a general pattern for yunnanolepidoids with additional characters distinguishing them from sinolepids and euantiarchs, such as a developed cranio-spinal process, an elongated endolymphatic duct, and a long occipital portion.

Discussion

In light of new data from Phymolepis and Yunnanolepis, we summarized the morphology on the visceral surface of head shield in antiarchs, and formulated an additional 10 characters for the phylogenetic analysis. These cranial characters exhibit a high degree of morphological disparity between major subgroups of antiarchs, and highlight the endocranial character evolution in antiarchs.

Introduction

Antiarchs, one of the most diverse and widespread fish groups during the Middle Paleozoic, have been resolved at the base of the diversification of jawed vertebrates in most of the recent phylogenetic studies (Brazeau, 2009; Davis, Finarelli & Coates, 2012; Zhu et al., 2012; Giles, Rücklin & Donoghue, 2013; Zhu et al., 2013; Dupret et al., 2014; Zhu, 2014; Giles, Friedman & Brazeau, 2015; Long et al., 2015; Qiao et al., 2016; Zhu et al., 2016). It is noteworthy that King et al. (2016) corroborated the placoderm monophyly and proposed antiarchs as a clade sister to petalichthyids and ptyctodonts based on the Bayesian tip-dated clock methods. Since the first description of antiarchs in 1840 (Eichwald, 1840), their general morphology has been well established (Young & Zhang, 1992; Janvier, 1996; Young & Zhang, 1996; Zhu, 1996; Lukševičs, 2001; Young, 2008; Young, 2010; Zhu et al., 2012; Long et al., 2015). However, the anatomical atlas of endocranium in antiarchs is poorly known, largely due to the absence of perichondral ossification (Denison, 1978). While the impressions on the overlying head shield help to restore the endocranial morphology to some extent (Stensiö, 1948; Denison, 1978; Moloshnikov, 2008), such as in Bothriolepididae (Stensiö, 1948; Young, 1984), Asterolepididae (Obruchev, 1933; Stensiö, 1969), and Minicrania lirouyii (Zhu & Janvier, 1996), little attention has been paid to character transformations of antiarch endocrania due to lack of corresponding data from primitive antiarchs.

Yunnanolepidoids are endemic antiarchs discovered from the South China and Indochina blocks (Liu, 1983; Pan & Dineley, 1988; Tông-Dzuy, Janvier & Phuong, 1996; Wang, Qu & Zhu, 2010). They are considered to be the most primitive antiarchs because of the absence of characteristic dermal brachial process (Chang, 1978; Zhang, 1978; Zhang, 1980; Young, 1981a; Long, 1983; Janvier, 1995; Zhu & Janvier, 1996; Carr, Johanson & Ritchie, 2009; Zhu et al., 2012), although their monophyly has not yet reached a consensus (Janvier & Pan, 1982; Young & Zhang, 1996). To date, yunnanolepidoids also represent the oldest known antiarchs, even though the date of the oldest yunnanolepidoid Shimenolepis (Wang, 1991) has recently been revised to Late Ludlow (Zhao et al., 2016).

Yunnanolepididae, a major clade of Yunnanolepidoidei, is characterized by the small brachial fossa and the crista transversalis interna posterior lying in front of the posterior process and pit of trunk shield. It includes the following seven genera: Yunnanolepis, Parayunnanolepis, Phymolepis, Mizia, Grammaspis, Chuanbeiolepis and Yunlongolepis (Chang, 1978; Zhang, 1978; Wang, 1988; Tông-Dzuy & Janvier, 1990; Zhu, 1996; Pan & Lu, 1997; Zhang, Wang & Wang, 2001; Pan et al., 2018).

Phymolepis Chang, 1978 (Figs. 1–3) is a yunnanolepid antiarch from the Lower Devonian of South China. The first description of the type species of Phymolepis, P. cuifengshanensis, was based on material (IVPP V4425) from the Xitun Formation of Cuifengshan in Qujing, Yunnan (Chang, 1978; Zhang, 1978). Amongst all the referred specimens of P. cuifengshanensis (Chang, 1978), V4425.7 (Fig. 4), a trunk shield with a row of medial marginal plates of pectoral fin, was later assigned to Yunnanolepis parvus (Zhang, 1980: pl. 5, fig. 1). V4425.7 differs from the holotype and other referred specimens of P. cuifengshanensis in its comparatively small size, a sharp median dorsal ridge running throughout the anterior median dorsal plate, and the absence of a conspicuous tergal angle of trunk shield. As such, we follow Zhang (1980) to remove V4425.7 from P. cuifengshanensis.

Figure 1 Holotype and paratype of Phymolepis cuifengshanensis.

(A–C) IVPP V4425.3, holotype, trunk shield in dorsal (A), ventral (B) and right lateral (C) views. (D–E) IVPP V4425.6, paratype, PMD in dorsal (D) and ventral (E) views. Abbreviations: aa, anterior angle of PMD; ADL, anterior dorsolateral plate; ala, anterolateral angle of PMD; AMD, anterior median dorsal plate; AVL, anterior ventrolateral plate; cf.AMD, area overlapping AMD; cf.PDL, area overlapping PDL; cr.tp, crista transversalis interna posterior; ddr, dorsal diagonal ridge of trunk shield; dl, dorsolateral ridge of trunk shield; dmr, dorsal median ridge; la, lateral angle of PMD; lc, main lateral line canal; lr, lateral ridge of lateral wall of trunk shield; MV, median ventral plate; pa, posterior angle of PMD; pda, posterior dorsal angle; PDL, posterior dorsolateral plate; pf, pectoral fossa; PL, posterior lateral plate; pla, posterolateral angle of PMD; plal, posterolateral angle of AMD; plr, posterior lateral ridge of PMD; PMD, posterior median dorsal plate; pr.p, posterior process of PMD; pt2, posterior ventral pit of dorsal wall of trunk shield; PVL, posterior ventrolateral plate; r.C, ridge caused by Chang’s apparatus; vl, ventrolateral ridge of trunk shield. Red arrow represents the direction of the specimen: a, anterior direction. Scale bar equals 1 cm.

Figure 2 Phymolepis cuifengshanensis (IVPP V4425.1).

(A) Dorsal view. (B) Ventral view. (C) Right lateral view. (D–E) Anterior view, photo (D) and drawing (E). Abbreviations: ADL, anterior dorsolateral plate; AMD, anterior median dorsal plate; AVL, anterior ventrolateral plate; cit, crista transversalis interna anterior; ddr, dorsal diagonal ridge of trunk shield; dma, tergal angle of trunk shield; dmr, dorsal median ridge of trunk shield; f.ca, fossa for neck-joint; iar, infra-articular ridge; lc, main lateral line canal; MV, median ventral plate; o.C, opening of Chang’s apparatus; pbl, postbranchial lamina; PDL, posterior dorsolateral plate; PL, posterior lateral plate; pnoa, postnuchal ornamented corner of ADL; PVL, posterior ventrolateral plate; r.C, ridge caused by Chang’s apparatus; sar, supra-articular ridge. Red arrow represents the direction of the specimen: a, anterior direction. Scale bars equal 1 cm.

Figure 3 Phymolepis cuifengshanensis (IVPP V4425.2).

(A) Dorsal view. (B) Ventral view. (C) Right lateral view. (D) Anterior view. Abbreviations: ADL, anterior dorsolateral plate; AMD, anterior median dorsal plate; AVL, anterior ventrolateral plate; ddr, dorsal diagonal ridge of trunk shield; dl, dorsolateral ridge of trunk shield; dma, tergal angle; dmr, dorsal median ridge; dtr, dorsal transverse ridge of trunk shield; La, lateral plate; lr, lateral ridge of lateral wall of trunk shield; MV, median ventral plate; Nu, nuchal plate; or, oblique ridge of lateral wall of trunk shield; orb, orbital fenestra; PDL, posterior dorsolateral plate; PL, posterior lateral plate; PM, postmarginal plate; pmc, postmarginal sensory canal; pnoa, postnuchal ornamented corner of ADL; PNu, paranuchal plate; PP, postpineal plate; PVL, posterior ventrolateral plate; SL, semilunar plate; vl, ventrolateral ridge of trunk shield. Scale bar equals 1 cm.

Figure 4 Yunnanolepis. parvus (IVPP V4425.7).

(A) Dorsal view. (B) Ventral view. (C) Anterior view. (D) Right lateral view. (E) Left lateral view. Abbreviations: ADL, anterior dorsolateral plate; AMD, anterior median dorsal plate; AVL, anterior ventrolateral plate; cit, crista transversalis interna anterior; dma, tergal angle; dmr, dorsal median ridge; pbl, postbranchial lamina; PDL, posterior dorsolateral plate; PL, posterior lateral plate; PMD, posterior median dorsal plate; p.pf, plates of pectoral fin; PVL, posterior ventrolateral plate. Red arrow represents the direction of the specimen: a, anterior direction; p, posterior direction. Scale bar equals 5 mm.

Young & Zhang (1996) described three specimens from the Xitun Formation (IVPP V9059.1–3) as P. cuifengshanensis, however these specimens are distinguishable from V4425.2 (Fig. 3), a referred specimen of P. cuifengshanensis preserving part of the head shield (Chang, 1978). The orbital fenestra in V9059.1–3 is semilunar in shape and occupies nearly half of the total breadth of the head shield. Accordingly, we also remove V9059.1–3 from P. cuifengshanensis.

Zhu (1996) assigned additional material from the Xishancun Formation at Liaokuoshan in Qujing (IVPP V10500.1–6, V10508.1–3) to P. cuifengshanensis, making the first occurrence of this genus no later than early Lochkovian. He also revealed the Chang’s apparatus (Zhu, 1996: fig. 11A) and the lateroventral fossa of trunk shield (Zhu, 1996: figs. 11F and 11G) in P. cuifengshanensis, and placed Phymolepis as the sister taxon of Mizia in the phylogenetic analysis.

To have a deeper understanding of yunnanolepidoids, particularly their cranial morphology, here we used high-resolution computed tomography (CT) to examine the internal morphology of dermal shield in P. cuifengshanensis. On the basis of resulting new data, P. cuifengshanensis was re-described in more detail. We also conducted the phylogenetic analysis of antiarchs based on a new character matrix expanded and revised from previous analyses (Zhu, 1996; Jia, Zhu & Zhao, 2010; Pan et al., 2018). Several cranial characters were compared and discussed among subgroups of antiarchs to illuminate the endocranial character transformations.

Materials and Methods

Material

The specimens of P. cuifengshanensis in this study are housed at the Institute of Vertebrate Paleontology and Paleoanthropology (IVPP), Chinese Academy of Sciences (CAS). The material were all collected from muddy limestone of the Xitun Formation in Cuifengshan, Qujing, Yunnan Province.

The Xishancun, Xitun, Guijiatun and Xujiachong formations (in ascending chronological order) represent the Early Devonian non-marine strata in Qujing District (Liu & Wang, 1973; P’an et al., 1978; Zhu, Wang & Fan, 1994; Liu, Gai & Zhu, 2018). The Xitun Formation consists mainly of grayish-green muddy limestone and mudstone (Fang et al., 1985; Xue, 2012), yielding a rich biota (Cuifengshan Assemblage) characterized by the diversification of sarcopterygians (Chang & Yu, 1981; Chang & Yu, 1984; Zhu, Yu & Janvier, 1999; Zhu, Yu & Ahlberg, 2001; Zhu & Yu, 2002) and primitive antiarchs including Yunnanolepis, Parayunnanolepis, Zhanjilepis, Chuchinolepis and Phymolepis. Other fishes in the Xitun Formation include the endemic agnathans (Liu et al., 2015), arthrodires (Dupret, Zhu & Wang, 2017), actinopterygians (Zhu et al., 2006; Lu et al., 2016), acanthodians and chondrichthyans, the latter two of which are mostly known from microvertebtrate remains (Wang, 1984). The Xitun Formation has been dated as late Lochkovian (c. 410–415 million years ago) with evidence from fossil assemblages (Gao, 1981; Fang et al., 1994; Zhao et al., 2011; Xue, 2012).

CT analysis

V4425.2, which preserved the only head shield material for Phymolepis in addition to the almost complete trunk shield, was CT scanned at the Key Laboratory of Vertebrate Evolution and Human Origins of Chinese Academy of Science, Beijing, using the 225 KV micro-tomography scanner (developed by Institute of High Energy Physics, CAS) with following parameters: 150 kV voltage; 100 mA current; 32.93 µm voxel size. All scans were conducted using a 720°rotation with a step size of 0.5°and an unfiltered aluminium reflection target. A total of 1,440 transmission images were reconstructed in a 2,048 × 2,048 matrix of 1,536 slices. The software Mimics v. 18.0 was applied for the three-dimensional reconstruction (segmentation and rendering).

Phylogenetic analyses

The character matrix of antiarchs herein consists of 42 ingroup taxa, two outgroup taxa (Kujdanowiaspis and Romundina), and 79 morphological characters. The matrix is modified from those of Zhu (1996), Jia, Zhu & Zhao (2010) and Pan et al. (2018), with revised codings and the addition of 10 cranial characters. More details including additional references and character re-formulations are provided in the Supplementary Information.

We performed a traditional search in TNT v 1.5 (Goloboff et al., 2008), using 1,000 random addition sequence replicates, saving 100 trees per replication. We assessed nodal supports through bootstrap values with 100 pseudoreplicates and Bremer decay indices. All characters were treated as equally weighted, and unordered (except Characters 19, 49 and 50). Character state transformations to the nodes of one of the most parsimonious trees (MPTs) were reconstructed in PAUP*4.0a (Swofford, 2003) adopting DELTRAN and ACCTRAN optimizations respectively. Character mapping was performed in MacClade 4.0 (Maddison & Maddison, 2000).

Results

Systematic paleontology

Placodermi McCoy, 1848	
Antiarcha Cope, 1885	
Yunnanolepidoidei Gross, 1965	
Yunnanolepididae Miles, 1968	
Phymolepis Chang, 1978	

Type species. Phymolepis cuifengshanensis Chang, 1978

Included species. Phymolepis guoruii Zhu, 1996

Emended diagnosis. Yunnanolepididae in which the posterior median dorsal plate bears a strong posterior process; the anterior median dorsal plate with anterior division longer than posterior division; anterior ventral process and pit situated just below a conspicuous tergal angle and at the same level of the lateral corners of the anterior median dorsal plate; sharp median dorsal ridge between the tergal and posterior dorsal angles.

Remarks. The diagnosis of this genus follows Zhu (1996) with a minor revision. While examining V4425.2 based on high-resolution CT, we noticed that the anterior ventral process and pit are situated at the level of the lateral corners of the anterior median dorsal plate rather than behind it.

PHYMOLEPIS CUIFENGSHANENSIS Chang, 1978	
(Figs. 1–3, 5–11 )	
1978 Phymolepis cuifengshanensis –Chang, p. 292, pl. XXV (5–7)	
1978 Phymolepis cuifengshanensis –Zhang, p. 147, figs. 10–12, pl. VI	
1996 Phymolepis cuifengshanensis –Zhu, p. 257, figs. 11–12, pls. I (8–10), IV (1–9)	

Holotype. IVPP V4425.3, a relatively complete trunk shield (Figs. 1A–Fig. 1C).

Paratype. IVPP V4425.6, a posterior median dorsal plate (Figs. 1D and 1E).

Referred specimens. IVPP V4425.1 (Fig. 2), trunk shield; V4425.2 (Fig. 3), nearly complete dermal shield only missing the anteriormost portion of head shield and the posterior median dorsal plate; V10500.1, V10508.1–3, posterior median dorsal plate; V10500.2, left anterior dorsolateral plate; V10500.3, left posterior dorsolateral plate; V10500.4, right posterior lateral plate; V10500.5, left anterior ventrolateral plate; V10500.6, right posterior ventrolateral plate.

Occurrence. The material was collected from two sites (Cuifengshan and Liaokuoshan) in Qujing city, eastern Yunnan, southwestern China.

Emended diagnosis. Phymolepis species in which the posterior process of the posterior median dorsal plate reaches one third of the plate length; the median dorsal ridge of trunk shield developed as a blunt elevation in front of the tergal angle and as a sharp crest behind the tergal angle.

Remarks. The diagnosis follows Zhu (1996) with an addition of the shape of the median dorsal ridge.

Description

Reconstruction and ornamentation

Using the complete specimen of Yunnanolepis chii (Zhang, 1978: V4423.101, fig.1) as a reference, Phymolepis cuifengshanensis could reach 84 mm in the dermal shield length, and represents the largest known species among Yunnanolepididae.

Figure 5 Outline restoration of the dermal shield of Phymolepis cuifengshanensis.

(A) Dorsal view. (B) Ventral view. (C) Right lateral view. Dashed lines delineate the unknown part. Scale bar equals 5 mm.

Figure 6 Life reconstruction of Phymolepis cuifengshanensis.

Artwork by Dinghua Yang.

Figure 7 Head shield of Phymolepis cuifengshanensis (IVPP V4425.2) based on high-resolution CT.

(A–B) Three-dimensional reconstructions in dorsal (A) and ventral (B) views. (C–D) Interpretative drawings in dorsal (C) and ventral (D) views. Abbreviations: a1, a2, attachment areas for the dermal operculum on the lateral and paranuchal plates, respectively; alr, anterior lateral ridge on head shield; c.csp, cavity for cranio-spinal process; cr.pm, paramarginal crista; cr.po, postorbital crista; cr.tv, transverse nuchal crista; d.end, opening for endolymphatic duct; d.sac?, depression for sacculus; dsc, depression caused by semicircular canal; ifc, infraorbital sensory canal; La, lateral plate; lc, main lateral line canal; mpl, middle pit-line; mr, median ridge of postpineal plate; nm, obtected nuchal margin; Nu, nuchal plate; occ, occipital cross commissure; oem, median occipital eminence; om, obstantic margin of head shield; ood, otico-occipital depression of head shield; orb, orbital fenestra; p.apo, anterior postorbital process; PM, postmarginal plate; pmc, postmarginal sensory canal; PNu, paranuchal plate; PP, postpineal plate; ppl, posterior pit-line; pp.th, postpineal thickening; proc, preobstantic corner of head shield; ptoc, postobstantic corner of paranuchal plate; sop, supraoccipital pit of head shield; sorb, suborbital fenestra. Scale bar equals 5 mm.

Figure 8 Cavities within the head shield of Phymolepis cuifengshanensis (IVPP V4425.2) based on high-resolution CT.

(A) Semi-transparent Nu in dorsal view. (B) Transparent Nu in anterior view. (C) Semi-transparent Nu in lateral view. (D–E) Semi-transparent right PNu in ventral (D) and posterior (E) views. (F) Right PNu in left lateral view. Abbreviations: cr.pm, paramarginal crista; cr.tv, transverse nuchal crista; c.vg, cavity for vagal process, d.end, opening for endolymphatic duct; Nu, nuchal plate; PNu, paranuchal plate. Red arrow represents the direction of the specimen; a, anterior direction; d, dorsal direction; l, left direction; v, ventral direction. Scale bar equals 2 mm.

Figure 9 Phymolepis cuifengshanensis. (IVPP V4425.2) based on high-resolution CT.

(A–C) Trunk shield in dorsal (A), ventral (B) and anterior (C) views. (D) Head and trunk shields in right lateral view. (E–F) AVL plates and their displaced fragments as preserved (E) and restored (F). Yellow dashed lines in (B) delimit restored portions. Abbreviations: ADL: anterior dorsolateral plate; AMD, anterior median dorsal plate; AVL, anterior ventrolateral plate; c.C; cavity of Chang’s apparatus; cg, caudal groove of trunk shield; cit, crista transversalis interna anterior; cr.tp, crista transversalis interna posterior; ddr, dorsal diagonal ridge of trunk shield; dma, tergal angle of trunk shield; dmr, dorsal median ridge of trunk shield; dtr, dorsal transverse ridge of trunk shield; f.ca, fossa for neck-joint; lc, main lateral line canal; lr, lateral ridge of lateral wall of trunk shield; MV, median ventral plate; Nu, nuchal plate; or, oblique ridge of lateral wall of trunk shield; PDL, posterior dorsolateral plate; PL, posterior lateral plate ; PM, postmarginal plate; pnoa, postnuchal ornamented corner of ADL; PP, postpineal plate; PVL, posterior ventrolateral plate; pbl, postbranchial lamina; rc, rostrocaudal canal; r.C, ridge caused by Chang’s apparatus; SL, semilunar plate. Red arrow represents the direction of the specimen: a, anterior direction; r, right direction. Scale bar equals to 5 mm.

Figure 10 Phymolepis cuifengshanensis (IVPP V4425.2) based on high-resolution CT.

(A) Head shield and anterior portion of trunk shield. (B) Axial section through the left AVL in CT slice, showing the positions of the internal cavity of Chang’s apparatus and the rostrocaudal canal. (C) Semi-transparent left ADL in ventral view. (D–E) Semi-transparent right AVL in dorsal (D) and lateral (E) views. (F) Semi-transparent left AVL in dorsal view. Abbreviations: ADL, anterior dorsolateral plate; AVL, anterior ventrolateral plate; c.C; cavity of Chang’s apparatus; cit, crista transversalis interna anterior; ifc, infraorbital sensory canal; lc, main lateral line canal; pbl, postbranchial lamina; rc, rostrocaudal canal. Red arrow represents the direction of the specimen: a, anterior direction; l, left direction; r, right direction. Scale bar equals 3 mm.

Figure 11 Phymolepis cuifengshanensis (IVPP V4425.2) based on high-resolution CT.

(A) AMD in ventral view. (B) Transverse section through the lateroventral fossa in CT slice. (C) Lateroventral fossa in internal view. (D) Transverse section through the left caudal groove in CT slice. (E) Axial section through the right caudal groove in CT slice. (F) Left caudal groove in lateral view. Abbreviations: AMD, anterior median dorsal plate; AVL, anterior ventrolateral plate; cf.ADL, area overlapping ADL; cf.PDL, area overlapping PDL; cg, caudal groove of trunk shield; cr.tp, crista transversalis interna posterior; f.lv, lateroventral fossa of trunk shield; lal, lateral angle of AMD; ms, median septum; PDL, posterior dorsolateral plate; PL, posterior lateral plate; plal, posterolateral angle of AMD; prv1, anterior ventral process of dorsal wall of trunk shield; pt1, anterior ventral pit of dorsal wall of trunk shield; PVL, posterior ventrolateral plate; wa, outer wall of caudal groove. Red arrow represents the direction of the specimen: a, anterior direction; d, dorsal direction; r, right direction. Scale bars equal 3 mm.

This re-investigation of P. cuifengshanensis brings together all referred specimens and leads to a new reconstruction (Figs. 5 and 6). The tentative restoration for the missing pre-orbital portion of head shield follows that of Y. chii (Zhang, 1978: fig. 1).

Small, round tubercles are densely distributed on the dorsal surface of the head and trunk shields. The tubercles are generally larger on various ridges and along outer margins of head shield than elsewhere. They are aligned parallel to the sutures between dermal plates, or radiated from the angles on the dorsal wall of trunk shield. In addition, they tend to form the rows along the sensory grooves. The tubercles on the lateral wall of the trunk shield are weakly developed and finer than those elsewhere. The ventral wall of the trunk shield is sparsely covered with tubercles that are slightly larger than the rest of the dermal shield.

Head shield

The orbital fenestra (orb, Figs. 7A and 7C) is comparatively small, and occupies about one fourth of the breadth of the head shield (Table 1). The obstantic margin (om, Fig. 7B) is straight and long, with the preobstantic corner of head shield (proc, Fig. 7A) at the midway of the postorbital division. The posterior margin of the head shield between well-marked postobstantic corners (ptoc, Figs. 7B and 7D), has convex lateral parts (formed mainly by the posterior margin of the paranuchal plate) and a slightly embayed mesial part (formed by the posterior margin of the nuchal plate). The obtected nuchal area (nm, Fig. 7A) occupies 27% length of the nuchal (Table 1).

Table 1 Measurements (in mm) and ratios for the head shield of Phymolepis cuifengshanensis (IVPP V4425.2).

Measurements were obtained by means of the digital visualization.

	Breadth	Length	Breadth/length	
Postpineal plate	6.72	2.49	2.67	
Nuchal plate	10.81	9.67	1.12	
Paranuchal plate	8.56	9.13	0.94	
Postmarginal plate	4.94	6.39	0.78	
Head shield (across preobstantic corners)	24.05			
Orbital fenestra	5.78			
Anterolateral margin of nuchal		3.23		
Posterolateral margin of nuchal		7.91		
Obtected nuchal area		2.65		
Breadth ratio between orbital fenestra and head shield	0.24	
Length ratio between posterolateral and anterolatereal margins of nuchal	2.45	
Length ratio between obtected nuchal area and nuchal	0.27	

The lateral plate (La, Fig. 7A) has long contact margins for surrounding dermal plates. In visceral view, the anterior attachment area for the submarginal plate is missing, however, the posterior attachment area on the lateral plate (a1, Fig. 7B and 7C) continues onto the postmarginal plate (a2, Figs. 7B and 7C).

The postpineal plate (PP, Fig. 7C) is wider than it is long. Its anterior margin is concave, unlike the straight margin in Parayunnanolepis. The postpineal thickening (pp.th, Fig. 7A) is extremely developed as a prominent tuberculate elevation, which totally encompasses the posterior border of the orbital fenestra and occupies about half of the postpineal plate length. On the visceral surface, paired postorbital cristae (cr.po, Fig. 7D) run somewhat obliquely along the anterior margin of the plate. Two cristae on either side of the postpineal plate are separated far away by a faint median ridge (mr, Fig. 7B), which lies at anterior margin of the plate and does not extend backwards as in many euantiarchs.

The nuchal plate (Nu, Fig. 7A) is broadest across the anterolateral angle. The postpineal notch is broad and deep. The posterolateral margin is about twice as long as the anterolateral one (Table 1). The robust anterolateral ridge (alr, Fig. 7A) sits mainly in the anterior division of the nuchal plate. The transverse nuchal crista (cr.tv, Fig. 7B) on the visceral surface is well developed and thickened laterally.

The paranuchal plate (PNu, Figs. 7A and 7B) is as broad as it is long. The obtected area of the plate is steeply inclined to the ornamented surface, especially near its suture with the nuchal plate. The postmarginal plate (PM, Fig. 7A) is rhombic and longitudinally extended. On the visceral surface, the attachment for the submarginal plate is narrower posteriorly.

Endocranium

Like other antiarchs, only the dorsal aspect of the endocranium can be inferred in P. cuifengshanensis from the impressions on the visceral surface of the head shield, which is digitally visible with a high level of details.

The otico-occipital depression of P. cuifengshanensis is deeper posteriorly, along with the gradually thickened paramarginal crista (cr. pm, Figs. 7B and 7D). This depression is laterally extended at the suture between the lateral and paranuchal plates, where the paramarginal crista lies underneath the infraorbital sensory groove. As such, the paramarginal crista in P. cuifengshanensis with the convex median part, differs from the straight one in asterolepidoids (Hemmings, 1978; Young, 1983) and the laterally concave one in bothriolepidoids (Young, 1988).

The anterolateral corner of the otico-occipital depression (p.apo, Fig. 7B), which represents the imprint for the anterior postobital process of endocranium (Young, 1984), is weakly developed and apically rounded. Significantly, it is located at the same level with the posterior border of the orbital notch.

Near the posterior end of the paramarginal crista, the paranuchal plate is deeply excavated by the large cavity (c.csp, Figs. 7B and 7D) for the cranio-spinal process of the endocranium. The cavity is conical and tapers laterally with a B/L ratio of around 3.0; its axis is perpendicular to the paramarginal crista (Figs. 8D–8F).

The semicircular depressions (dsc, Fig. 7D) sit just in front of the level of cranio-spinal processes. The anterior and posterior semicircular depressions are relatively short, and meet in a confluence that is located midway between the posterior border of the orbital notch and the transverse nuchal crista. As the lateral extension of the otico-occipital depression roughly levels with the confluence, this lateral extension appears to relate with the labyrinth cavity (d.sac?, Fig. 7D) as seen in Arenipiscis westolli (Young, 1981b: fig. 6). In view of the otic region, which can be estimated by the position of semicircular depressions, lies mainly in the anterior half of the otico-occipital depression, so the occipital region of P. cuifengshanensis is fairly long compared with euantiarchs.

Median to the posterior end of the semicircular depression, the internal pore for the endolymphatic duct (d.end, Figs. 7B and 7D) is rounded and situated far ahead of the transverse nuchal crista, while the external pore (d.end, Fig. 7A) is situated far posteriorly at the anterior margin of obtected nuchal area. The distance between the internal pores of both sides is 2.5 times longer than the distance between the external ones. The digital visualization reveals that the endolymphatic duct of P. cuifengshanensis is a long and roughly straight tube. It runs posterodorsally within the nuchal plate, swings laterally while close to the midline of the plate and opens to the exterior (Figs. 8A–8C).

Posteriorly, a pair of supraoccipital pits (sop, Figs. 7B and 7D) is positioned just in front of the transverse nuchal crista. This pit is easily distinguished from the internal pore of the endolymphatic duct by its large size and ellipsoidal shape. The pit is dorsomedially oriented within the nuchal plate, and gradually tapers off just beneath the ornamented surface (Figs. 8A–8C). The supraoccipital pit also occurs in Vukhuclepis (Racheboeuf et al., 2006: fig. 4) and Yunnanolepis at the same position. It is noteworthy that Liu (1963): Fig. 1) misidentified the supraoccipital pit in Yunnanolepis as the internal pore for the endolymphatic duct.

Just anterior to the cavity for the cranio-spinal process, a corner (c.vg, Fig. 8F) in a nearly right angle is set on approximately at the posterior end of a semicircular depression level, and thus the hindmost level of the otic region. This corner is also positioned between the anterior postorbital process and cranio-spinal process of endocranium. Therefore, we tentatively interpret this corner as the depression of the vagal process as it shares the same topological relationships to that of arthrodires and petalichthyids.

Trunk shield

The trunk shield is fairly high, with a conspicuous tergal angle (Figs. 2C–2D, 9C–9D) taking up almost half of the trunk shield height. The small pectoral fossa (pf, Fig. 1C) is set just above the bottom of the trunk shield, and occupies a quarter of the lateral wall height (Table 2). Both the dorsolateral and ventrolateral ridges of the trunk shield are robust (dl, vl, Figs. 1C and 3C).

Table 2 Measurements (in mm) and ratios for the trunk shield of Phymolepis cuifengshanensis.

Specimen number	V4425.1	V4425.2	V4425.3	
Dorsal wall of trunk shield (excluding posterior median dorsal plate)	Breadth (B)	23.0	22.0	39.0	
Length (L)	31.0	34.0	50.0	
B/L	0.7	0.7	0.8	
Lateral wall of trunk shield	Length	30.0	34.0	40.0	
Height (H)	10.0	15.0	20.0	
L/H	3.0	2.3	2.0	
Ventral wall of trunk shield	Breadth	28.0	30	45.0	
Length	34.0	48.0	–	
B/L	0.8	0.6	–	
Anterior median dorsal plate	Breadth	18.0	22.0	28.0	
Length	26.0	32.0	41.0	
B/L	0.7	0.7	0.7	
Anterior dorsolateral plate	Breadth	13.0	14.0	17.0	
Length	17.0	20.0	26.0	
Height	6.0	8.0	11.0	
B/L	0.8	0.7	0.7	
L/H	2.8	2.5	2.4	
Anterior ventrolateral plate	Breadth	15.0	20.0	–	
Length	19.5	29.0	–	
B/L	0.8	0.7	–	
Posterior dorsolateral plate	Breadth	8.0	9.5	13.0	
Length	15.5	17.0	28.0	
Height	–	4.0	8.0	
B/L	0.5	0.6	0.5	
L/H	–	4.3	3.5	
Posterior lateral plate	Length	17.0	19.0	25.0	
Height	7.0	8.0	10.0	
L/H	2.4	2.4	2.5	
Posterior ventrolateral plate	Breadth	18.0	20.0	–	
Length	16.0	18.0	–	
Height	7.0	8.0	–	
B/L	1.1	1.1	–	
L/H	2.3	2.3	–	
Semilunar plate	Breadth	–	8.0	–	
Length	–	4.0	–	
B/L	–	2.0	–	
Median ventral plate	Breadth	13.0	–	23.0	
Length	14.0	–	24.0	
B/L	0.9	–	1.0	

The dorsal wall has a convex anterior margin. The median dorsal, dorsal diagonal and dorsal transverse ridges (dmr, ddr, dtr, Figs. 1–3) on the dorsal wall radiate from the tergal angle as in Mizia longhuaensis, Yunnanolepis porifera and Chuchinolepis qujingensis (Zhu, 1996: figs. 4, 5C–5D, 21A). The lateral wall carries the lateral and oblique ridges (lr, or, Figs. 1C, 3C and 9D), which are widely developed in yunnanolepidoids.

The main lateral line (lc, Figs. 2C and 10A) runs posteriorly very close and subparallel to the dorsolateral ridge of trunk shield. It terminates at the end of the dorsolateral ridge on the posterior dorsolateral plate.

The anterior median dorsal plate (AMD, Figs. 1–3A, 9A and 11A) is pentagonal in shape. The posterolateral margin is embayed near its posterior end as in Yunnanolepis (Zhang, 1980: fig. 3B). The concave posterior margin of the plate is delimited laterally by distinct posterolateral angles (plal, Figs. 1A and 11A). The tergal angle lies at the same level with the lateral corner of the plate. Internally, the anterior ventral pit (pt1, Fig. 11A) with thin rim is located right beneath the tergal angle. It extends posteriorly to form a low ridge (prv1, Fig. 11A).

The posterior median dorsal plate (PMD, Figs. 1D and 1E) bears a large posterior process (pr.p, Fig. 1E), which occupies about two fifths of the plate in length and three fourths in breadth. The dorsal median ridge and the posterior lateral ridges of both sides (plr, Fig. 1D) converge to the posterior dorsal angle, which is developed as a small nodule. The posterior corner of the plate (pa, Fig. 1D) is rounded. Internally, the crista transversalis interna posterior (cr.tp, Fig.1E) is developed as a low ridge just anterior to the posterior ventral process (pt2, Fig. 1E).

The anterior dorsolateral plate (ADL, Figs. 2C, 3C and 9D) consists of articular, dorsal and lateral laminae. The transversely extending articular fossa (f.ca, Figs. 2D and 9C) is delimited by the supra- and infra-articular ridges (sar, iar, Fig. 2E). The supra-articular ridge, which extends laterally from the postnuchal ornamented corner (pnoa, Figs. 2C , 3C and 9C), is longer than the infra-articular one. The dorsal lamina is slightly arched with a dorsal diagonal ridge. The dorsal division of the ridge caused by Chang’s apparatus (r.C, Figs. 2E and 9D) is positioned on the lateral lamina adjacent to the obstantic margin of the head shield.

The anterior ventrolateral plate (AVL, Fig. 9) consists of lateral and ventral laminae, which meet at the ventrolateral ridge. The lateral lamina bears the ventral division of the ridge caused by Chang’s apparatus near its anterior margin. The ventral lamina shows a shallow semilunar notch. Internally, both the posterior branchial lamina (pbl, Figs. 2D–2E and 10D) and crista transversalis interna anterior (cit, Figs. 2D–2E and 10D) are strongly developed. The posterior branchial lamina, ornamented by denticulate ridges, is present close to the anterior margin of the trunk shield. It runs anteromedially from the lateral lamina of the plate to the ventral lamina as a narrow band. The crista transversalis interna anterior is located immediately behind the postbranchial lamina. Dorsally, the crista extends from the AVL to the base of the articular fossa on the ADL (Figs. 2D, 2E and 9C), where it is just behind Chang’s apparatus (c.C, Fig. 10C). The left AVL overlaps the right.

Between the posterior branchial lamina and crista transversalis interna anterior, a canal (rc, Figs. 10A, 10D–10F) is present just anterior to the internal cavity of Chang’s apparatus (Fig. 10B). The canal passes ventrally along the lateral wall of the trunk shield. With a relatively large diameter, it probably carries both vessels and nerves and corresponds to the rostrocaudal canal in Chuchinolepis (Young & Zhang, 1992), sinolepids and euantiarchs, which is similarly positioned to supply the fin muscles (Young, 2008).

The posterior dorsolateral plate (PDL, Figs. 2C and 3C) consists of the dorsal and lateral laminae. The dorsal lamina is slightly less than twice as long as it is broad (Table 2).

The posterior lateral plate (PL, Figs. 2C, 3C and 9D) is arched along the lateral ridge of trunk shield. The anteroventral margin is concave, and longer than the anterodorsal one. The dorsal margin of the PL overlaps the PDL.

The posterior ventrolateral plate (PVL, Figs. 2C and 9B) consists of lateral and ventral laminae. The subanal division of the ventral lamina is too short to define. The left PVL overlaps the right one.

On the visceral surface of the trunk shield, a fossa (f.lv, Figs. 11B and 11C) is located at the thickened junction of the AVL, PVL and PL plates, as in Yunnanolepis and Zhanjilepis (Zhu, 1996). The fossa was termed the ‘lateroventral fossa’, and regarded as a synapomorphy of yunnanolepids by Zhu (1996).

Posteriorly, a deeply grooved internal structure (cg, Fig. 11E) is developed along the caudal opening of trunk shield. The groove has a smooth internal surface, delimited anteriorly and posteriorly by the developed crista transversalis interna posterior and posterior margins of trunk plates (PVL, PL and PDL) respectively. It consists of upper and lower halves divided by a thin septum (ms, Figs. 11D 11F). The similar structure in Yunnanolepis porifera, as well as in Pterichthyodes milleri (Hemmings, 1978: fig. 15D), was assumed to be related to internal fertilization (Long et al., 2015).

The semilunar plate (SL, Fig. 9B) is triangular in shape, and is approximately twice as broad as it is long. It is overlapped posteriorly by the AVL. Internally, the postbranchial lamina extends anteromesially from the AVL onto the semilunar plate, and meets the lamina from the opposite.

The median ventral plate (MV, Figs. 1B, 2B and 9B) is rhombic. The exposed surface accounts for two fifths of the ventral wall of trunk shield in length and a half of the ventral wall in breadth. The plate is thinner than the surrounding plates.

Discussion

Anatomical comparisons of several cranial characters in antiarchs

The restoration of the endocranium in antiarchs was mainly based on the imprints of its dorsal aspect on the visceral surface of the head shield (Stensiö, 1948; Stensiö, 1969; Miles, 1971; Denison, 1978; Young, 1984). The exception was Minicrania, which preserved the internal cast of the endocranial canals and part of the cranial cavity, thus providing information on its deeper endocranial structures (Zhu & Janvier, 1996).

In yunnanolepidoids, the visceral surface of head shield was known in Yunnanolepis (Liu, 1963: fig. 1) and Chuchinolepis (Tông-Dzuy & Janvier, 1990: fig. 17). The digital visualization of Phymolepis shows not only its visceral surface of head shield but also some internal architecture within the dermal plates, such as the trajectory of the endolymphatic duct and the cavity for the cranio-spinal process. We also re-examine the holotype (V2690.1, Fig. 12A) and one referred specimen (V4423.3, Fig. 12B) of Y. chii from the Early Devonian of Qujing, and provide more details for the visceral surface of head shield in Yunnanolepis. Based on these new data, we make comparisons in antiarchs, and show a high degree of morphological disparity with respect to the endocranium.

Figure 12 Head shields of Yunnanolepis chii in visceral view.

(A) IVPP V2690.1. (B) IVPP V4423.3. Abbreviations: c.csp, cavity for cranio-spinal process; cr.pm, paramarginal crista; cr.po, postorbital crista; cr.tv, transverse nuchal crista; d.end, opening for endolymphatic duct; d.sac?, depression for sacculus; dsc, depression caused by semicircular canal; fm, unpaired insertion fossa on head shield for levator muscles; mr, medial ridge of postpineal plate; ood, otico-occipital depression of head shield; p.apo, anterior postorbital process; r.spr, subpremedian ridge; sop, supraoccipital pit of head shield; tlg, transverse lateral groove of head shield. Scale bars equal to 5 mm.

Anterior postorbital process. The endocrania of gnathostomes share a developed lateral projection where the orbits meet the otic capsules (Brazeau & Friedman, 2014). This process was termed the ‘anterior postorbital process’ in placoderms, and deemed as supporting the hyoid arch articulation and delimiting the posterior boundary of the spiracular chamber by its anterior surface (Young & Zhang, 1996; Brazeau & Friedman, 2014). The positional differences of the anterior postorbital process (and associated cranial nerves) along the longitudinal axis of dermal shield were thought to be informative for phylogenetic analysis (Carr, Johanson & Ritchie, 2009; Dupret et al., 2017).

Antiarchs have a well-developed anterior postorbital process. The process extending in front of the anterior border of the orbital notch, has been considered as one of the synapomorphies uniting Bothriolepis and Grossilepis (Zhang & Young, 1992; Zhu, 1996; Jia, Zhu & Zhao, 2010; Pan et al., 2018). Accordingly, the process behind the anterior border of the orbital notch is referred to a plesiomorphy of antiarchs and this state has been simply summarized as “anterior postorbital process short” in previous phylogenetic analyses (Zhang & Young, 1992).

When examining the short anterior postorbital process in antiarchs, we recognized that this state can be subdivided into two conditions: the anterior postorbital process at the same level with the posterior border of the orbital notch in yunnanolepidoids, Minicrania (Zhu & Janvier, 1996), and probably Sinolepis (Liu & P’an, 1958; Long, 1983); the process anteriorly beyond the posterior border of the orbital notch, but behind its anterior border in euantiarchs excluding Grossilepis and Bothriolepis. In this case, the distinction between these two conditions can be added to the transformation series of the anterior postorbital process.

Postorbital crista. The postorbital crista in antiarchs separates the otico-occipital depression from the orbital region in front. In several euantiarchs, the crista extends obliquely from the lateral plate to the nuchal plate as a mesial wall of the semicircular depression, such as in Bothriolepis (Stensiö, 1948), Monarolepis (Young & Gorter, 1981), Pterichthyodes (Hemmings, 1978) and Wufengshania (Pan et al., 2018). For the rest of antiarchs including yunnanolepidoids, the postorbital crista runs from the lateral plate to the postpineal plate rather than the nuchal plate as a transversely directed crest embracing the suborbital fenestra posteriorly.

Supraotic thickening. The supraotic thickening (Young, 1983: sot, fig. 3D) is bounded posteriorly by the transverse nuchal crista and extensively developed at its connection with the crista. As the supraotic thickening is porous, different from the rest of dermal skeleton in microstructure, it was considered as a junction that is co-ossified with both the endocranium and overlying head shield (Stensiö, 1948; Karatajūte-Talimaa, 1963; Moloshnikov, 2004; Moloshnikov, 2008). Euantiarchs have a persistent supraotic thickening with the exception of Microbrachius, which bears a deep groove throughout the whole length of the otico-occipital depression (Hemmings, 1978: figs. 25C–F). The presence of this thickening on the visceral surface of euantiarchs is in stark contrast to the condition in yunnanolepidoids, Minicrania and sinolepids, which lack any thickening in the corresponding area.

Median occipital crista. This crista was first identified and named by Stensiö (1931): cro, figs. 11 and 12) in Bothriolepis, and was also termed the ‘posterior median process’ by Hemmings (1978) in Pterichthyodes. Lying on the descending lamina of occipital part of the head shield, it is separated from the otico-occipital depression by the transverse nuchal crista in euantiarchs. A shallow depression of levator muscles (termed the ‘insertion fossa on head shield for levator muscles’) usually flanks on each side of the crista. In sinolepids, such as Grenfellaspis (Ritchie et al., 1992), the insertion fossa is elongated as that in euantiarchs but lacks the median crista. In yunnanolepidoids, the insertion fossa is either very short (Yunnanolepis, Fig. 12B), or totally absent (Phymolepis, Fig. 7B).

Posterior process of head shield. The posterior process of head shield (prnm, see Young (1988): figs. 7B, 37C and 44A) in euantiarchs, also termed the ‘nuchal process’ or ‘posterior median process’ (Long & Werdelin, 1986; Moloshnikov, 2004; Moloshnikov, 2008; Moloshnikov, 2010), was first identified and named by Stensiö (1931: figs. 4, 9 and 12). Although the median occipital crista is usually continuous with the posterior process of head shield, the process is apparently independent of the crista in development as evidenced by Asterolepis and Remigolepis, which possess the process but lack the crista. The process is usually developed in euantiarchs, in contrast to its absence in yunnanolepidoids and sinolepids.

Among non-antiarch placoderms, the posterior process is also known in petalichthyids (Liu, 1991; Pan et al., 2015) and arthrodires (Wang & Wang, 1983; Gardiner & Miles, 1990; Young, 2005; Carr & Hlavin, 2010; Rücklin, Long & Trinajstic, 2015).

The cavity for cranio-spinal process. The cranio-spinal process was named by Nielsen (1942), and also termed the ‘supravagal process’ by Stensiö (1969) and the ‘paroccipital process’ by Eaton (1939). It is widely developed in early gnathostomes, including arthrodires (Young, 1979), petalichthyids (Stensiö, 1925), acanthodians (Miles, 1973), actinopterygians (Patterson, 1975) and dipnoans (Miles, 1977). However, the cavity for cranio-spinal process on the visceral surface of head shield, which might function for fixing the endocranium to the external bony shield, is only found in primitive antiarchs and some arthrodires.

The cranio-spinal process in yunnanolepidoids is strongly developed, as indicated by the large cavity for the process. The process and the corresponding cavity in euantiarchs were either reduced or absent (Young, 1984).

Supraoccipital pit. The supraoccipital pit of the head shield is present for housing the endocranial supraocciptial process. It is bounded posteriorly by the transverse nuchal crista in yunnanolepidoids. The same condition is also seen in Grenfellaspis (Ritchie et al., 1992) and Minicrania, despite the supraoccipital pit in the latter has ever been interpreted as impression of endolymphatic sac (Zhu & Janvier, 1996; Dupret et al., 2017). In euantiarchs, the supraoccipital pit is only seen in few Bothriolepis species with different positions: either immediately anterior to the transverse nuchal crista as exemplified by B. tatongensis (Long & Werdelin, 1986), or on the transverse nuchal crista as in B. macphersoni and B. portalensis (Young, 1988).

In non-antiarch placoderms, the supraoccipital pit has been observed in petalichthyids (Liu, 1991: cv.ifNu, fig. 2), and most arthrodires, including Holonematidae (Miles, 1971: tf; figs. 53 and 117; Young, 2005: if.pt, fig. 2C), Buchanosteidae (Young, 1981b: if.pt, fig. 6), Coccosteoidea (Miles & Westoll, 1968: p.pts.Nu, fig. 2A), Dunkleosteoidea (Zhu, Zhu & Wang, 2016: f.pt.u, fig. 5; Carr & Hlavin, 2010: pt.u, fig. 6A) and Dinichthyidae (Carr & Hlavin, 2010: pt.u, fig. 1A).

Trajectory of endolymphatic duct. The trajectory of the endolymphatic duct through the dermal bone in respect of length and orientation mainly depends on the relative position between the internal and external pores. This character was considered informative for the resolution of placoderm interrelationships. The trajectory had ever been decomposed into two states: vertical (a trait in most non-arthrodire placoderms), long and oblique (a trait shared by arthrodires) by Goujet & Young (1995). Coates & Sequeira (1998) considered the posteriorly oriented duct as a primitive character of gnathostomes as it is shared by agnathans, placoderms and osteichthyans. Brazeau (2009) suggested the presence of posterodorsally angled trajectory as an arthrodire character and the absence of oblique trajectory of endolymphatic duct as a character shared by antiarchs, Brindabellaspis and petalichthyids. Our study herein shows the condition in antiarchs is more complicated than previously thought.

In antiarchs, the distance between the internal pores is usually greater than that of external ones (Stensiö, 1948; Karatajūte-Talimaa, 1966; Long, 1983), and the endolymphatic duct extends dorsomesially. As the external pore of endolymphatic duct is always positioned close to the posterior edge of the nuchal plate in antiarchs, the relative position of the internal pore along the antero-posterior axis reflects the relative length and orientation of the endolymphatic duct.

In yunnanolepidoids, the internal pore of endolymphatic duct is located far in front of the transverse nuchal crista, and thus far from the external pore. As such, the endolymphatic duct is elongated through the nuchal plate and obliquely oriented. Sinolepids (Ritchie et al., 1992; Janvier, 1996) and euantiarchs differ in having a short, slight oblique endolymphatic duct as the internal pore is positioned just anterior to the external one.

In non-antiarch placoderms, the elongated endolymphatic duct is also present in arthrodires (Young, 2010; Dupret et al., 2017). However, the endolymphatic duct of arthrodires is directed dorsolaterally, not dorsomesially as in antiarchs.

Occipital portion of endocranium. The internal pore for the endolymphatic duct in antiarchs, is located roughly at the posterior boundary of the semicircular depression on the visceral surface of head shield as that in arthrodires (Zhu, Zhu & Wang, 2016), hence we use this pore as a proxy to denote the otic-occipital boundary. Taking the length of the otic-occipital depression as the constant variable in antiarchs, the length between the internal pore of endolymphatic duct and the posterior border of otic-occipital depression represents the occipital proportion in endocranium.

In yunnanolepidoids, the internal pore on the visceral surface of the head shield is far from the transverse nuchal crista as described above, implying the occipital portion of the endocranium is elongated as in arthrodires (Zhu, Zhu & Wang, 2016). By comparison, the occipital portion is short in other antiarchs (Stensiö, 1948; Ritchie et al., 1992).

Confluence of anterior and posterior semicircular canals. The anterior and posterior semicircular canals meet at the confluence in the medial part of the inner ear (Dupret et al., 2017). As the posterior border of orbital notch and the transverse nuchal crsita roughly border the anterior and posterior margins of the otic-occipital depression respectively, we can use them as references to estimate the relative position of the confluence in endocranium.

In yunnanolepids and Minicrania (Zhu, 1996), the confluence is halfway between the posterior border of the orbital notch and the transverse nuchal crista. By comparison, the confluence is closer to the transverse nuchal crista than to the posterior border of orbital notch in sinolepids and euantiarchs.

Phylogenetic results

The maximum parsimony analysis yields 726 MPTs of 179 steps each (consistency index = 0.4693; retention index = 0.8045). All the MPTs are summarized as a strict consensus tree (Fig. 13A) and a 50% majority-rule consensus tree (Fig. 13B). One MPT is selected to illustrate the character transformations at nodes (Fig. 14A), and the list of synapomorphies defining various nodes is shown in Supplementary Information.

Antiarchs (Fig. 14A: node 1) are characterized by up to 10 synapomorphies including two newly proposed cranial features (Character 270, absence of posterior process of head shield; Character 381, presence of supraoccipital pit). Character 27 shows a reversal in euantiarchs (Fig. 14A: node 15). Character 38 is a highly homoplastic character, and shows a reversal in euantiarchs and a parallelism in Bothriolepis.

Yunnanolepidoids (Fig. 14A: node 2) form the basal members of antiarchs, consistent with the position as they were first phylogenetically analysed (Zhu, 1996). However in new scenario, Chuchinolepis, Vanchienolepis and a clade formed by yunnanolepids, Zhanjilepis and Heteroyunnanolepis (Wang, 1994) fall into a polytomy with remaining antiarchs. In yunnanolepids, Yunnanolepis is the sister group of a polytomic clade comprising Phymolepis, Mizia and Parayunnanolepis.

Figure 13 Phylogenetic results of antiarchs based on a revised data matrix.

(A) Strict consensus tree of 726 parsimonious trees (tree length = 179, consistency index = 0.4693, homoplasy index = 0.5307, retention index = 0.8045, rescaled consistency index = 0.3775). Numbers above and below nodes represent bootstrap values (≥50% are shown) and Bremer decay indices, respectively. (B) 50% majority-rule consensus tree of 726 parsimonious trees based on the same dataset as in (A). Numbers next to nodes indicate the percentage of the shortest trees in which the partition is supported (100% are not shown).

Figure 14 Phylogenetic results of antiarchs and visceral surface conditions of head shield among major antiarch subgroups.

(A) One of the most parsimonious trees with node numbers defining various clades. Named nodes: 1, Antiarcha; 2, Yunnanolepidoidei; 11, Sinolepididae; 15, Euantiarcha; 16, Microbrachiidae; 26, Asterolepidoidei. (B) Restorations of the head shields in ventral view to show endocranial character transformations, redrawn from Ritchie et al. (1992), Stensiö (1948), Karatajūte-Talimaa (1963) and Young (1983). Vertical bars on the right side show the longitudinal proportion of otico-occipital region of endocranium on the head shied (blue region), the location of the confluence of semicircular canals (orange circle), the location of the internal pore for endolymphatic duct (purple circle). Abbreviations: c.csp, cavity for cranio-spinal process; cr.im, inframarginal crista; cr.tv, transverse nuchal crista; cro, median occipital crista; d.end, opening for endolymphatic duct; dsc, depression caused by semicircular canal; f.cu, cucullaris fossa; fm, unpaired insertion fossa on head shield for levator muscles; p.apo, anterior postorbital process; prnm, posterior process of head shield; sop, supraoccipital pit of head shield; sot, supraotic thickening of head shield; tlg, transverse lateral groove of head shield.

Four new cranial characters provide further support the monophyly of euantiarchs (Fig. 14A: node 15), including one uniquely shared character (Character 361, anterior postorbital process lying in front of posterior level of orbital notch) and three homoplastic characters (Character 261, presence of median occipital crista of head shield; Character 271, presence of posterior process of head shield and Character 380, absence of supraoccipital pit of head shield).

Microbrachiids (Fig. 14A: node 16) are resolved as the sister group of the remaining euantiarchs, and the conventional bothriolepidoids are resolved as a paraphyletic assemblage. These results are congruent with previous analyses of Zhu (1996) and Pan et al. (2018). Relationships of the remaining bothriolepidoids (Fig. 14A: node 19) are completely unresolved in the strict consensus tree, which may be related to the large number of missing data in some of them. Euantiarchs excluding microbrachiids bear one uniquely shared endocranial character (Character 251, presence of supraotic thickening of head shield).

Our analysis that incorporate new cranial characters yields resultant trees, which are consistent with previous resolutions of Zhu (1996), Jia, Zhu & Zhao (2010) and Pan et al. (2018) in broad phylogenetic pattern. Under the new phylogenetic scenario, we can trace the character transformations relating to the dorsal aspect of endocranium in antiarchs.

Yunnanolepidoids (Figs. 7 and 12 and 14A ) and Minicrania show primitive character states, such as the anterior postorbital process being posteriorly positioned (Character 360), presence of cranio-spinal process (Character 371) and supraoccipital process (Character 381), anterior and posterior semicircular canals being anteriorly positioned (Character 390), long endolymphatic duct (Character 400), and long occipital portion (Character 410).

At the node comprising sinolepids and euantiarchs (Fig. 14A: node 10), there are two derived endocranial character states: short endolymphatic duct (Character 401) and short occipital region (Character 411). Euantiarchs differ from sinolepids in possessing the following derived states: the anterior postorbital process lying in front of the posterior level of orbital notch (Character 361), and the absence of the supraoccipital process (Character 380). In short, there exists a large morphological disparity relating to the dorsal aspect of endocranium between yunnanolepidoids, sinolepids and euantiarchs.

Conclusions

The re-investigation of Phymolepis cuifengshanensis with the assistance of high-resolution CT scanning offers comprehensive information for this taxon and new insights into the morphology and phylogeny of antiarchs.

The exoskeleton of Phymolepis cuifengshanensis shows typical yunnanolepid characters, such as the small orbital fenestra, presence of both developed postbranchial lamina and crista transversalis interna anterior on the trunk shield. The endocranium of P. cuifengshanensis also resembles that of other yunnanolepidoids in the presence of developed cranio-spinal process and supraoccipital process, the anterior postorbital process lying at the same level with the posterior border of the orbital notch, elongated endolymphatic duct and long occipital region.

We compare cranial characters among subgroups of antiarchs, and formulate 10 additional characters that deemed to be of phylogenetic significance. Phylogenetic analysis of a revised and expanded dataset draws new perspectives on the interrelationships of antiarchs, and corroborates the monophyly of yunnanolepidoids by the presence of cavity for cranio-spinal process.

The character transformations relating to the dorsal aspect of endocranium in antiarchs are inferred under the new phylogenetic scenario. By comparison to yunnanolepidoids and Minicrania, which retain several primitive endocranial traits, sinolepids and euantiarchs evolved two apomorphic features (short endolymphatic duct and short occipital portion). Euantiarchs are more derived in the anterior postorbital process lying in front of the posterior level of orbital notch, and the absence of the supraoccipital process.

Supplemental Information

Supplemental Information 1 Supplementary text

Phylogenetic analysis: (1) Character list. (2) Taxa and principal sources of data. (3) Data matrix with 79 morphological characters for 44 taxa. (4) Characters and character states defining major clades shown in Fig. S1. (5) Supplementary references.

Click here for additional data file.

Supplemental Information 2 Character matrix of antiarchs

Nexus file comprising 44 taxa and 78 morphological characters.

Click here for additional data file.

Figure S1 Supplementary figure to show character transformations in antiarchs

Unambiguous character state changes mapped across one of the most parsimonious trees (Fig. 14).

Click here for additional data file.

Video S1 Phymolepis cuifengshanensis (IVPP V4425.2) based on high-resolution CT

Continuous transverse sections of the specimen.

Click here for additional data file.

Video S2 Head and trunk shields of Phymolepis cuifengshanensis

Three-dimensional model of IVPP V4425.2.

Click here for additional data file.

Video S3 Head shield of Phymolepis cuifengshanensis (IVPP V4425.2) based on high-resolution CT

Three-dimensional semi-transparent model to show the cavities and endolymphatic ducts within the dermal shield.

Click here for additional data file.

We thank Dinghua Yang for the life restoration, Xiaocong Guo for suggestions on the interpretative drawings, You-an Zhu for discussions on arthrodire characters, and Liantao Jia and Wei Gao for the assistance in taking photographs.

Additional Information and Declarations

Competing Interests

Author Contributions

Data Availability

The authors declare there are no competing interests.

Yajing Wang performed the experiments, analyzed the data, contributed reagents/materials/analysis tools, prepared figures and/or tables, authored or reviewed drafts of the paper, approved the final draft.

Min Zhu conceived and designed the experiments, performed the experiments, analyzed the data, contributed reagents/materials/analysis tools, prepared figures and/or tables, authored or reviewed drafts of the paper, approved the final draft.

The following information was supplied regarding data availability:

Dryad DOI: 10.5061/dryad.k1g46v2.

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
