# Peer review of "Redescription of Phymolepis cuifengshanensis (Antiarcha: Yunnanolepididae) using high-resolution computed tomography and new insights into anatomical details of the endocranium in antiarchs"

_PeerJ, doi:10.7717/peerj.4808_

## Round 0.1 · original submission · Minor Revisions

This is a nice contribution which reveals additional anatomical details of Phymolepis from the Early Devonian of China and nicely illustrates the importance of CT scanning in revealing additional information on the endocranium of antiarchs. The manuscript is in a good state and I would like to see it published. There are however some minor, but crucial points which I would like to see addressed for publication:

Title: It might be appropriate to slightly change atlas to anatomical details of the endocranium in the title (see comments by reviewer 2).

Measurements: It would be good to make plate measurements and indices available in one or multiple tables (see comments by reviewer 2).

Descriptions: The descriptions could be streamlined and shortened a bit (see Reviewer 2).

CT-Scanning Reproducibility: I welcome to use of tomography which has revolutionized our field and is also crucial for your paper. It would however be appropriate to make additional information available for scientific verification and reproducibility (Davies et al. 2017). Following Davies et al. 2017 you would need to provide at least a full-resolution image stack (e.g., TIFF) as well as the final 3D-models (e.g., stl format) used in the study. Furthermore, additional information on the scanner would be necessary. You report the current, voltage, voxel size, but you should also report additional details on the CT-scanner (type: brand or custom), number of projections, exposure time and filter thickness (if any).

Please take into account the additional suggestions by the reviewers in addition to these points.

Suggested reference:

Davies TG, Rahman IA, Lautenschlager S, Cunningham JA, Asher RJ, Barrett PM, Bates KT, Bengtson S, Benson RBJ, Boyer DM, Braga J, Bright JA, Claessens LPAM, Cox PG, Dong X-P, Evans AR, Falkingham PL, Friedman M, Garwood RJ, Goswami A, Hutchinson JR, Jeffery NS, Johanson Z, Lebrun R, Martínez-Pérez C, Marugán-Lobón J, O'Higgins PM, Metscher B, Orliac M, Rowe TB, Rücklin M, Sánchez-Villagra MR, Shubin NH, Smith SY, Starck JM, Stringer C, Summers AP, Sutton MD, Walsh SA, Weisbecker V, Witmer LM, Wroe S, Yin Z, Rayfield EJ, and Donoghue PCJ. 2017. Open data and digital morphology. Proceedings of the Royal Society B: Biological Sciences 284.

Reviewer 1 ·

Basic reporting

This manuscript is an excellent redescription of Phymolepis, a rare and poorly known antiarch from the Lower Devonian of Yunnan, China. The work is perfectly executed and provides a number of new anatomical information which allow to refine previous phylogenetic analyses of this important group of placoderm fishes. I have no major correction to make. Just a small comment concerning Vanchienolepis, from the Lower Devonian of Vietnam: Remember that the skull roof referred to Vanchienolepis langsonensis by Tong-Dzuy Thanh & Janvier (1990, fig.14) was found isolated from any typical Vanchienolepis thoracic armour, and there is no guarantee Yunnae that is belongs to this genus. Its attribution to Vanchienolepis is only based on its small size and the fact that it was found in a block which also contained undoubted Vanchienolepis plates. However, it is thus not ruled out that it is in fact a skull roof of a small, juvenile Yunnanolepis.
Minor corrections:
L. 64: yes, there is no information about the actual neurocranium of antiarchs, except perhaps the posterior part of the brain cavity in Minicrania (Zhu & Janvier 1996). I suspect that, like in osteostracans and galeaspids, the endocranial structures are more often preserved in very small individuals, possibly because of the proximity of osteogenic cells recruited from the overlying dermal skeleton.
L. 77: although their monophyly has not yet reached a consensus…
L.327: …within the nuchal plate, swings laterally when close to the midline…
L.353: Yunnanolepis
L.629 and 639: homoplastic is the original term, but homoplasious is also correct.
The quality of the illustration is outstandings.

Experimental design

not applicable

Validity of the findings

This description provides new key information about the anatomy of yunnanolepidoid antiarch fishes. Very useful in the framework of the question the origin of jawed vertebrates

Additional comments

This manuscript is an excellent redescription of Phymolepis, a rare and poorly known antiarch from the Lower Devonian of Yunnan, China. The work is perfectly executed and provides a number of new anatomical information which allow to refine previous phylogenetic analyses of this important group of placoderm fishes. I have no major correction to make. Just a small comment concerning Vanchienolepis, from the Lower Devonian of Vietnam: Remember that the skull roof referred to Vanchienolepis langsonensis by Tong-Dzuy Thanh & Janvier (1990, fig.14) was found isolated from any typical Vanchienolepis thoracic armour, and there is no guarantee Yunnae that is belongs to this genus. Its attribution to Vanchienolepis is only based on its small size and the fact that it was found in a block which also contained undoubted Vanchienolepis plates. However, it is thus not ruled out that it is in fact a skull roof of a small, juvenile Yunnanolepis.
Minor corrections:
L. 64: yes, there is no information about the actual neurocranium of antiarchs, except perhaps the posterior part of the brain cavity in Minicrania (Zhu & Janvier 1996). I suspect that, like in osteostracans and galeaspids, the endocranial structures are more often preserved in very small individuals, possibly because of the proximity of osteogenic cells recruited from the overlying dermal skeleton.
L. 77: although their monophyly has not yet reached a consensus…
L.327: …within the nuchal plate, swings laterally when close to the midline…
L.353: Yunnanolepis
L.629 and 639: homoplastic is the original term, but homoplasious is also correct.
The quality of the illustration is outstandings.

·

Basic reporting

The paper is a detailed description of an important taxon of fossil antiarch placoderm from the Early Devonian of China, with new data revealed through CT scanning, and will be of much interest to workers in this field. However the text can be shortened in places, as we no longer require for every plate to be described in minute detail (similar to Miles 1968 Bothriolepis paper) when figures can now do this clearly. The descriptive text should only used to highlight key main differences in anatomy of the taxon in question as compared with closely related taxa that might be confused with it. Many statements in the descriptions are unnecessary as they state facts that either apply to all antiarchs or to all yunanolepidoids, so can be deleted (these are listed below for lines 233-286 for the headshield only, but this must be applied to entire descriptive section). The figures are all excellent and show the anatomy very clearly. The discussion and phylogenetic sections are is clear and require little corrective editing.
Indices for plate proportions is of little value if the sample is 1 or 2 specimens eg L356 says lateral wall of Tr/sh has a L/H ration of 2-3. This means there is 50% variation in the ratio, so it is actually taxonomically useful to include here? I would recommend all the plate measurements and indices for all specimens (only 11 listed- L197-204) be all listed in a table or several tables laid out in a similar style to how Miles & Dennis (1979) and Dennis & Miles (1979-1984).
The paper would thus benefit from a major rewrite of the descriptive text as it stands and revising how mensuration data is presented.

Specific points to address:-
L59-61 excludes King et al 2016, a major contribution to placoderm systematics that disagrees with the opening statement of this paper as it did not find antiarchs at the base of all placoderms, so a statement about this paper needs be included.
L64 'atlas' is not right word here as little is known of antiarch enodcrania, they are all hypothetically restored. I think "anatomical details of the endocranium" would be better.
L83 "crista transversalis interna posterior" etc throughout the text should be in italics as in Stensio 1948, Zhu 1996 etc.

Description
L236 “suborbital fenestra is smaller than orbital fenestra” is the same in all antiarchs, so not a useful statement. Delete or qualify by adding comparisons to other related yunanolepuidoid antiarchs where it is different, so the statement is more meaningful.
L240 –index of 2.16 i to 2 decimal places means little, with such small sample size, simplify to 2.1, but should be listed in a table I(as I’ve said above).
L244 “relatively long” compared to what else?. Obstantic margins are only long if the headshield is long in its posterior half-so all yunanolepidoids have this trait. Statements like this need be qualified or they add nothing of value to the description. Qualify or delete.
L249 “elongated’ as above-relative to what taxa? Qualify or delete.
L250-254 these 2 sentences apply to nearly all antiarchs as far as I know.. Clarify why this needs to be stated here, because it differs in xx closely-related species, or delete.
L256- postmarginal groove short-as in all yunnanolepidoids? Qualify with more comparisons, or delete.
L257 “relatively broad” relative to what taxa? Qualify or delete.
L258 “relatively long” relative to what taxa? Qualify or delete.
L261-B/L index of 1.5 is close enough (one decimal point given such small sample size).
L270 should be ‘anterolateral’ not lateral angles
L271-272: Unnecessary stamen as the nuchal is divided into these 2 halves in all antiarch nuchal plates. Delete.
L275-7: statement about “the X shaped pit lines…” in front of obtected nuchal area applies in all antiarchs where well-preserved, eg its in Bothriolepis (e.g. Stensio 1948 vol 1 text fig 10) plus Gogo specimens. Delete.
L277-278: “..as seen in other yunnanolepidoids” means it add nothing to the description that is either unique to the taxon being described, or adds new information about yunnanolepidoids, so qualify with other comparisons or delete.
L278-280 about “the transverse nuchal crista reaching maximum thickness etc” is unnecessary as it applies to all antiarchs. Delete.
You can see from my comments above that the same kind of detailed editing needs be applied for the entire part of the descriptive section to L446, making more use of the figures. A table needs be included to states indices and list all specimens from which the measurements were taken.
No need to write out the indices in the description or state if the plate is broad or relatively long etc. The suggested new table listing measurements and indices will do this. Just draw attention to how it differs from other closely related taxa and refer to the figure. This will greatly tighten up the descriptive part of the MS.

Other description comments:
L315 'Arenipiscis' is misspelt.
L320 Add ‘the’ before ‘semicircular depression”
L346-4476 need be edited carefully along same lines s my comments for headshield. Cut out unnecessary statements that apply to all antiarchs or all yunanolepidoids, qualify all statements about being relatively long, broad, elongate, or robust as for how these apply to other related taxa.
L348 please add a label for the ‘pectoral fossa’ in the figures
Discussion
L507 Not Miles 1968 but should be Hemmings 1978 for Microbrachius?
L543 Long & Trinajstic 2015 not in refs and it actually doesn’y exist, so please clarify which paper is meant here. Trinajstic et al 2015 Biological Reviews 90 paper perhaps? Or Long & Trinajstic 2010 Annual Reviews in Earth and Planetary Sciences paper perhaps?

Phylogenetics etc
L687 “short occipital division” in sinolepids and euantiarchs: or is this a by product of the elongation of the premedian area, making the occiput seem shorter? In essence the occiput area is still long in sinolepidoids, but the premedian plate is also longer than in yunanaolepidoids. The occipital area is only shortened relative to entire postorbital area of the headshield in Bothriolepidoids and Asterolepidoids and certain taxa on their stem as shown in your final figures. Worth adding a short sentence discussing this.

Figures
All excellent. Perhaps revise Fig 14 to include the entire head shield of Grenfellaspis in Fig 14B ventral view as all the head shield plates of this taxon are shown in internal view in the paper, so it could easily be reconstructed.

Experimental design

All fine, see my comments above though regarding layout and presentation of mensuration data. Research is meaningful and well defined. Investigation is rigorous, methods OK.

Validity of the findings

validity and interpretation of the findings is good. My only issue was with the way measurement indices are presented -needs to be in a table showing all speciemens measured, as some indices are based on few specimens, so this needs be clarified to be statistically significant.

Additional comments

This is very good work, but the descriptions follow an older style of presentation that can be greatly tightened up by editing down the descriptive section, letting the excellent figures do the talking. I hope my comments will improve the MS and make it clearer for readers to use.

---

## Round 0.2 · Minor Revisions

Thank you for making these changes. Your paper is as good as accepted.

I just had one additional question concerning the availability of your CT data. Thank you for sharing various videos. I understand your position that you don´t want to share all data before it has been been used for additional studies. However, various platforms exist to publish (e.g., MorphoMuseum) or store CT-data (http://morphomuseum.com/links). At least some of these platforms allow to embargo data (e.g., morphosource; www.morphosource.org) meaning that nobody other than yourself (and collaborators) would have access to it within this embargo time. I just wanted you to consider it one more time (or at least for future reference) as it aids scientific reproducibility (e.g., Davies et al. 2017) .

Thank your for your understanding.

---

## Round 0.3 · accepted · Accept

Thank you for your willingness to make these files available for scientific reproducibility through dryad. Please make sure to make these files you made available through dropbox become available through dryad upon publication.The correct doi number should also be mentioned in your manuscript.

#